# Feasibility, Safety, and Effects of an Aerobic Training Program with Blood Flow Restriction on Functional Capacity, and Symptomatology in Women with Fibromyalgia: A Pilot Study

**DOI:** 10.3390/biomedicines12081895

**Published:** 2024-08-19

**Authors:** José Carlos Rodríguez-Bautista, Guillermo López-Lluch, Patricia Rodríguez-Torres, Álvaro López-Moral, Jesús Quijada-Carrera, Javier Bueno-Antequera, Manuel Blanco-Suárez, Óscar Cáceres-Calle, Diego Munguia-Izquierdo

**Affiliations:** 1Physical Performance and Sports Research Center, Department of Sports and Computer Science, Section of Physical Education and Sports, Faculty of Sport Sciences, Universidad Pablo de Olavide, 41013 Seville, Spain; alopmor1@alu.upo.es (Á.L.-M.); jbueant@upo.es (J.B.-A.); dmunizq@upo.es (D.M.-I.); 2Department of Physiology, Anatomy and Cell Biology, Andalusian Centre of Developmental Biology (CABD-UPO-JA), Centro de Investigación en Rendimiento Físico y Deportivo (CIRFD), Universidad Pablo de Olavide, 41013 Seville, Spain; glopllu@upo.es; 3Department of Internal Medicine, Hospital Universitario Nuestra Señora de Valme, 41014 Seville, Spain; patricia.rtorres@hotmail.com; 4Rheumatology Department, Hospital Viamed Santa Ángela de la Cruz, 41014 Seville, Spain; jesusquijada.carrera@gmail.com; 5SHC Medical, Hospital Viamed Santa Ángela de la Cruz, 41014 Seville, Spain; manuelblancosuarez@gmail.com (M.B.-S.); ocaceres@telefonica.net (Ó.C.-C.)

**Keywords:** fibromyalgia, blood flow restriction, functional capacity, feasibility, safety

## Abstract

Background: Evidence suggests that aerobic training with blood flow restriction is beneficial for treating fibromyalgia. This study evaluated the feasibility, safety, and effects of an aerobic training program with blood flow restriction for women with fibromyalgia. Methods: Thirty-seven women with fibromyalgia were included, and thirteen with an average age of 59 ± 3, a BMI of 26 ± 3, and who were polymedicated started the intervention period. The intervention group performed aerobic exercise with blood flow restriction using occlusive bands placed in the upper part of the rectus femoris, with a total duration of 14 min of restriction divided into two periods of 7 min with a rest period of 3 min and a total session duration of 17 min. Pressure intensity was measured using the visual pain scale (VAS), scoring 7 out of 10 (n = 7). The non-intervention group performed aerobic exercise without restriction of blood flow for the same periods, rest periods, and total duration of the session (n = 6). The intervention included 2 weekly sessions with 72 h between aerobic walking for 9 weeks. Walking was measured individually using the rating of perceived exertion scale (RPE) with an intensity between 6 and 7 out of 10. Visual and verbal support for the VAS and RPE scale was always provided throughout the sessions supervised by the investigator. Functional capacity was assessed using tests (six-minute walk test, incremental shuttle walk test, knee extension and handgrip test by dynamometer, 30 s chair stand test, and timed up-and-go test). Symptomatology was assessed using questionnaires (Widespread Pain Index, Symptom Severity Score, Fibromyalgia Impact Questionnaire, and Multidimensional Fatigue Inventory), and blood samples were collected. Results: There were no adverse effects, and only one participant in the intervention group withdrew. Between-group and intragroup differences showed that the intervention group obtained improvements in the functional tests; CST *p* = 0.005; 6MWT *p* = 0.011; Handgrip *p* = 0.002; TUGT *p* = 0.002 with reduced impact of the disease according to the questionnaires; FIQ Stiffness *p* = 0.027 compared with the nonintervention group. Biochemical results remained within normal ranges in both groups. Conclusions: Blood flow-restricted aerobic training may be feasible, safe, and more effective than unrestricted aerobic training as a physical exercise prescription tool to improve cardiorespiratory fitness, strength, balance, and stiffness in women with fibromyalgia.

## 1. Introduction

Fibromyalgia is a chronic disease that causes widespread pain and extreme exhaustion. It is more common in women and affects, on average, 2.10% of the world’s population, 2.31% of the European population, and 2.40% of the Spanish population [1]. Fibromyalgia is characterized by diffuse, wide, and varied symptomatology, with direct involvement of central and peripheral physiological parameters, with a high prevalence of neuropathic pain and small-fiber neuropathic pain symptoms [2,3]. These limitations severely affect patients with fibromyalgia in terms of functional autonomy and quality of life, largely due to limitations in their physical condition [4] that manifest in generalized fatigue, lack of tolerance to physical exercise [5], and late, insufficient, and ineffective post-exertional recovery due to exacerbation of exertion and perceived pain [6]. These circumstances result in depression and psychological deterioration [7], low adherence to various physical exercise programs [8,9], and an inability to obtain the potential benefits of aerobic exercise in the medium and long term [10].

The context of fibromyalgia patients makes it difficult to maintain their daily life activities essentially because of a neuromuscular limitation causing a low rate of force production, greater deterioration compared with healthy people [11], and poor cardiorespiratory condition [12]. These limitations cause patients with fibromyalgia constant overexertion and, consequently, generalized decline; therefore, it is of utmost importance to provide a response through a physical intervention that can provide an adaptable, safe, and useful tool as an adjuvant element that improves these patients’ symptoms and functional capacity.

Currently, treatments for patients with fibromyalgia vary depending on the severity of symptoms, concomitant elements of the disease, and its limitations in terms of mental health, pain, sleep quality, physical autonomy, and degree of disease involvement. According to the most current guidelines based on scientific evidence, the following are considered to be the most recommendable: pharmacological therapy divided into analgesic, anticonvulsant, antidepressants, and muscle relaxants; and non-pharmacological therapy, such as aerobic physical exercise, strengthening exercise, behavioral–cognitive therapy, occupational therapy, hydrotherapy, and mindfulness [13].

There is now sufficient research to support aerobic exercise because it seems to reduce pain perception, depression, and mental and physical health [14], although with the problem of low adherence to exercise programs [8,9]. Patients with fibromyalgia could benefit from aerobic training adapted to their specific needs. There are currently no studies that have conducted aerobic training with blood flow restriction in patients with fibromyalgia. This study is the first to conduct an investigation with an intervention period using this tool. Occlusive training has several characteristics that make its use in fibromyalgia possible. It limits the time and intensity of exercise required to achieve improvements. This aspect is fundamental for such a physically limited population [15]. People with fibromyalgia need more time for recovery [16], do not tolerate intense exercise adequately [17], and their tolerance to training programs is limited [9].

Occlusive training can generate sufficient stimuli through acute blood flow restriction (BFR) to produce significant improvements in healthy and clinical populations [18,19,20,21,22,23,24,25,26]. Elements to highlight in a fibromyalgia-focused blood flow restriction aerobic training program include the ability to significantly reduce the volume and intensity of the applied training load necessary to produce adaptations that may result in better control of physical and mental fatigue from physical exercise. Avoid the exacerbation of perceived pain during and after training sessions with increased recovery time. All these factors can result in a better rate of adherence to exercise with blood flow restriction, allowing adaptations in aerobic and neuromuscular capacity at endurance and strength levels, which can translate into greater distance covered with less exertion and perceived pain [16,19], improvements in aerobic endurance with effects on the quality and decrease of muscle mass loss in older adults [20,21], evaluation of safety parameters without muscle damage or alteration of biomarkers [22,23], increased maximum oxygen consumption (VO2max) with only 2 weeks of training in trained athletes [24], improvements in lipid and hematological profiles in overweight people, compatibility with metabolic syndrome [25], and improvements in autonomy and walking performance in elderly people with knee osteoarthritis [26]. Therefore, considering the limitations and needs of patients with fibromyalgia, aerobic exercise together with blood flow restriction may be a tool to implement in their daily routine to improve their functional autonomy.

The objective of this pilot study was to evaluate the feasibility, safety, and effects of an intervention based on a low-intensity, low-volume aerobic training program with blood flow restriction with respect to functional capacity and the impact on symptomatology in women with fibromyalgia. The hypothesis is that exercise through blood flow restriction in women with fibromyalgia is an effective, safe, and feasible tool to consider in the prescription of physical exercise to improve functional capacity and reduce symptoms.

## 2. Materials and Methods

This study followed the Transparent Reporting of Evaluations with Nonrandomized Designs (TREND) (Appendix A [27]). The protocol was registered with ClinicalTrials.gov (identifier NCT06001034) and approved by the ethics committees of the Virgen Macarena and Virgen del Rocío University Hospitals (0560-N-22) Junta de Andalucía, Government of Spain. All subjects were duly informed about the phases of the research, objectives, and protocol in the informed consent form they signed. They did not receive compensation for their participation.

The proportion of women diagnosed with fibromyalgia in Andalusia, Spain, according to the National Institute of Statistics [28], has a prevalence of approximately 12.35%, indicating a population of approximately 34,000 women. The calculated N: Z_a_^2^ = 1.96^2^ (if the confidence level is 95%); *p* = expected proportion (in this case 5% = 0.05); q = 1 − *p* (in this case 1 − 0.05 = 0.95); d = precision (in this case 7%), N = 37.

The study was conducted at the private Hospital Viamed Santa Ángela de la Cruz in collaboration with the public Hospital Universitario Nuestra Señora de Valme in Seville, Spain. The recruitment process was conducted over 8 weeks by members of the internal medicine and rheumatology team as well as by the research team of the Pablo de Olavide University in the fibromyalgia associations of the province of Seville.

Internal medicine specialists selected subjects who met the criteria for participation. The subjects were between 53 and 64 years old, included using standardized questionnaires on the impact of fibromyalgia with a minimum score according to the American College of Rheumatology (ACR) 2010 [29], and had a positive diagnosis of fibromyalgia by the medical team. Subjects with cardiovascular disease were excluded from the study because of direct contraindications to the study, failure to exceed the minimum score according to ACR 2010, and limiting motor disease.

The initial evaluation included measures of functional capacity, disease effects, and blood sampling. The final evaluation was performed after 9 weeks of intervention, including the same tests performed previously. Before the initial blood draw, blood pressure measurements were taken in a seated position and averaged over 2 shots in the same arm to establish the intervention group (aerobic exercise with BFR) with blood pressures no higher than 130/80 mmHg overall and the nonintervention group (aerobic exercise without BFR) with blood pressures no higher than 135/85 mmHg overall or in isolation. All subjects who exceeded 135/85 mmHg blood pressure as a whole or in isolation were discarded from the study because participation criteria included being neither diagnosed with hypertension nor treated with antihypertensives [30,31].

The choice of the groups (Figure 1) was strictly based on ethical and safety considerations, as there was no previous study of this population, the side/adverse effects of aerobic training programs with blood flow restrictions in the clinical population at the musculoskeletal and vascular level [32], or possible interactions of the drugs prescribed in the last 12 months (Table 1) focused on analgesia, non-steroidal anti-inflammatory drugs (NSAIDs), and antidepressants, concerning the participants’ physiological parameters [32,33,34,35]. Therefore, the best blood pressure was used as the basis for selecting the AE-BFR and AE groups. Likewise, Table 2 shows that there were no baseline differences between the groups at the start of the study, except for the group selection criteria variable. This decision was established based on factors that could influence the increase in blood pressure during the session (cuff pressure, medication, poor physiological state of the patient with fibromyalgia) [12,30,31,32,33,34,35,36].

The intervention began one week after the baseline assessment. A code was assigned to each participant to blind the data in the statistical analysis. The results of the group assignment were unknown until the patients accepted or declined to participate in the study. All measurements were assessed by researchers blinded to group assignment. All participants were assessed by the same researchers to minimize interexaminer error. The same exercise physiologist with several years of experience in researching people with fibromyalgia conducted both group exercise sessions. Due to the nature of the study, participants allocated to the aerobic exercise group with blood flow restriction used a tool placed on the lower body that exerted direct pressure on the thigh that did not allow blinding.

### 2.1. Measures

Feasibility was assessed based on the following [37,38]:
Self-management: the subject’s ability to perform the training and manage its relative intensity autonomously with the visual analog scale (VAS) and rate of perceived exertion (RPE);Attendance: number of exercise sessions the participant completed;Dropout: number of participants who completed the treatment.

Safety was assessed based on the following [22,23,39]:
No abnormal biochemical findings or exacerbations of pain or perceived exertion;No alterations in oxygen saturation (SpO_2_) or heart rate (HR).

Functional capacity [40] was assessed using cardiorespiratory fitness, strength, and balance tests [41,42,43,44,45,46].

Cardiorespiratory fitness was assessed by the distance walked (without running or jogging) in the six-minute walk test (6MWT) [41] (UpoTV—6-Minute Walk) and incremental shuttle walk test (ISWT) [42,43] (UpoTV—Incremental Shuttle Walk Test).

Balance was assessed using the timed up-and-go test (TUGT) [44] in an upright position on a chair supported by a wall with a backrest, covering a distance of 3 m marked with a cone as measured with an odometer (digital wheel 320 mm, PrimeMatik, USA) with an accuracy of ±0.01 m; at the signal (ready, set, go), the patient had to get up, cover the marked distance, return, and sit down as quickly as possible, without jogging or running.

Strength was assessed by a handgrip test [45,46] using a dynamometer (TKK 5401 Grip-D, Takei, Tokyo, Japan. 2022) with an accuracy of ±2.0-kg force (kgf). Participants in a standing position were tasked with squeezing the handgrip as quickly and as hard as possible for 5 s with the elbow in full extension. The test was conducted twice with each hand, each having a 1 min break between squeezes, and the maximum value of the two best attempts of each hand was chosen. Individual differences were measured by relative grip strength (grip strength divided by body mass). Knee extension strength (KES) [47] was determined using a manual muscle tester equipped with a Lafayette Dynamometer (Model 01163, Lafayette Instrument Company, Lafayette, IN, USA) with an accuracy of ±0.09 kgf at 90° on both legs, fixed. The test was performed twice on each leg for 5 s with a 1 min rest between each, and the maximum value of the 2 best attempts of both was selected. To account for individual differences, the relative knee extension force (extension force divided by body mass) was used. Finally, strength was evaluated with the 30 s chair stand test (CST) [48] (UpoTV—30 s chair stand) in an upright position using a chair with a backrest leaning against the wall. At the signal (ready, set, go), with arms crossed and feet resting on the floor, the participants stood up and sat down continuously for 30 s.

Symptomatology was assessed by fibromyalgia impact questionnaires using the Widespread Pain Index (WPI) (range 0–19) and Symptom Severity Score (Sscore) (range 0–12). The diagnostic criteria for fibromyalgia are satisfied if the following three conditions are met: (1) WPI equal to or greater than seven and SSscore equal to or greater than five, or WPI between three and six plus SSscore equal to or greater than nine; (2) symptomatology was maintained at similar levels for the last three months; (3) the fibromyalgia sufferer did not have another disease that could explain the pain [29,49]. The Fibromyalgia Impact Questionnaire Spanish version (FIQ) is a brief 10-item, self-administered instrument that measures physical functioning, work, depression, anxiety, morning tiredness, pain, stiffness, fatigue, good feeling, and absence from work (not applicable). The total FIQ score is between 0 and 100, as each item has a score of 0–10 once adapted. Thus, 0 represents the highest functional capacity and quality of life, and 100 the worst [50]. The Multidimensional Fatigue Inventory, Spanish version (MFI), is a 20-item scale designed to evaluate five dimensions of fatigue: general fatigue, physical fatigue, reduced motivation, reduced activity, and mental fatigue [51].

Weight and height were recorded using a medical scale with a (RGT-200, Baxtran, Spain) measuring rod with an accuracy of ±100 g, followed by blood samples (heparinized and nonheparinized) from fasting participants. The samples were extracted by venipuncture with a butterfly safety needle supported by a vacuum tube, centrifuged in a 5810 r Eppendorf centrifuge for 15 min at 3000× *g* at room temperature, and stored in a chest at −80 °C until analysis. Malondialdehyde (MDA) was analyzed using a Cayman TBARS Assay kit 10009055 with an absorbance measurement plate reader at 530 nm, and total antioxidant capacity (TAC) was analyzed using a Sigma-Aldrich CS0790 kit with absorbance measurement at 405 nm. Colorimetric analysis of both kits was performed using a Cytation 3 multimodal reader (BioTek, Winooski, VT, USA).

Coenzyme Q-10 was extracted according to a protocol [52] using a Beckman 166–126 system equipped with a 15-cm Kromasil C-18 column and analyzed using ultraviolet-based detectors (System Gold 168) and a Coulochem III electrochemical detector (ESA, Washington, DC, USA). Biochemical analysis was performed using Roche test strips and calibration strips by automatic photometry analysis (Reflotrom, Boheringer Mannhein, Stuttgart, Germany).

The biochemical variables analyzed included the following lipid profiles: total cholesterol (Chol), high-density lipoproteins (HDL), low-density lipoprotein (LDL), and triglycerides (TG); total antioxidant capacity (TAC); antioxidant coenzyme Q-10: total coenzyme Q-10 (CoQ10). Ratio: CoQ10/HDL, CoQ10/LDL, CoQ10/Chol, and CoQ10/TG. As a lipid peroxidation marker, malondialdehyde (MDA), creatine kinase (CK), and glucose. Transaminases: gamma-glutamyl transferase (GGT), glutamate-pyruvate transaminase (GPT), glutamic-oxaloacetic transaminase (GOT), and alkaline phosphatase (ALP). Renal variables: urea, creatinine (Cr), and uric acid (Uric ac). All samples were stored and analyzed at the Centro Andaluz de Biología del Desarrollo (CABD), Universidad Pablo de Olavide (Seville).

### 2.2. Intervention

The intervention was conducted over the course of 9 weeks, twice a week, with 72 h between sessions, which were always in the morning, in the Hospital Santa Ángela de la Cruz, Seville. Prior to this, a 3-session familiarization period was conducted with an occlusive tool to ensure that the subjects were autonomous when setting the pressure using the VAS as a visual reference. Likewise, we familiarized participants with the intensities to be performed using the RPE scale with visual support, previous stride recording, and the pedometer recalibration process.

During the intervention period, data were recorded before and after each session for HR and SpO_2_ using a pulse oximeter (1905 RI-FOX N, GmbH Riester, Jungingen, Germany) with an accuracy of ±3%, as well as on the VAS, followed by RPE 0–10 and distance traveled with records taken at the end (Table 3 and Table 4). Each session had a duration of 14 min of activity: 7 min of walking followed by 3 min of rest (2 sitting and 1 standing) in occlusion at 0 mmHg (that would increase when standing), with VAS applied to the last 7 min of walking. Visual and verbal support for the VAS and RPE scale was always provided throughout the sessions supervised by the investigator.

For occlusion training, bands (VBM 20-54-528, Medizintechnik, GmbH, Sulz am Neckar, Germany) were used with a manual inflator and a pressure gauge (VBM20-18-601-VBM20-18-602, Medizintechnik, GmbH, Sulz am Neckar, Germany). Occlusive bands were placed on the upper part of the rectus femoris. The occlusion pressure was measured using a restriction protocol with VAS in the standing position [53], and it was progressively increased until reaching a value of 7 out of 10 on this scale. Likewise, the intensity of aerobic training was measured by the RPE CR-10 scale of perceived exertion [54], keeping the subject’s intensity between 6 and 7 out of 10. The distance covered was recorded with a pedometer (HJ-320-E Omron Healthcare Co., Ltd., Kyoto, Japan) with a 3-axis sensor and accuracy of ±5%, recalibrated at the beginning of each session with each subject for stride-length adjustment with a 10-step measurement protocol (distance traveled in 10 steps in centimeters divided per number of steps).

### 2.3. Statistical Analysis

Variables were calculated using descriptive statistics. Data distribution was checked for normality using the Shapiro–Wilk test. Variables with skewed distributions were logarithmically transformed to obtain normal distributions. Paired Student’s *t*-test, qualitative differences, and standardized differences (±90% confidence level) determined the relative changes within each group. Paired Student’s *t*-test, qualitative differences, and standardized differences (±90% confidence level) were calculated to compare baseline variables. To determine differences between groups, a multiple 2 × 2 (group × time) mixed ANOVA was performed, adjusting for multiple comparisons using the Bonferroni method by dividing the 0.050 significance level by the number of comparisons. In addition, qualitative and standardized differences (±90% confidence level) were calculated. The effect size for standardized differences was determined using Cohen’s d statistic, and the magnitude of effect sizes using the Hopkins scale, where 0 to 0.19 = insignificant, 0.20 to 0.59 = small, 0.60 to 1.19 = moderate, and 1.2 to 1.99 = large, >2 = very large [55]. The probability of a true difference between groups was classified qualitatively as almost certainly not: <0.5%; extremely unlikely: 0.5% to 5%; unlikely: 5% to 25%; possible: 25% to 75%; likely: 75% to 95%; very likely: 95% to 99.5%; almost certain: >99.5%. A significant effect was defined as greater than 75% [56]. A significant difference was assumed when the difference between subjects was ≥0.2 SD. Qualitative assessment shows the likelihood of significant differences between groups, referring to possible differences, probable differences, very likely differences, and almost certain differences. The tests were performed using SPSS Statistics version 22.0 for Windows (IBM Corp., Armonk, NY, USA). Post hoc power analysis was performed using SPSS versión 29 (IBM SPSS, IBM Corp.) based on the effect size and sample size, assuming an alpha of 0.05. An adequate power analysis was defined as ≥0.8.

## 3. Results

The cuff pressure throughout the intervention period did not exceed 115 mmHg of occlusion at any time (mean ± SD = 105.8 ± 6.71) to keep the VAS score below 7. The attendance rate at the 18 sessions during the 9 weeks of training was 85.7% in the aerobic exercise group with blood flow restriction (AE-BFR) and 100% in the aerobic exercise group (AE), with only one dropout (14.3%; 1/7) in the AE-BFR group due to repeated absence for personal reasons (Appendix A).

### 3.1. Feasibility and Safety

Substantial differences were shown between groups in the distance covered per session (Table 3) and in HR after the sessions (Table 4), with the AE-BFR group obtaining a greater increase in distance covered and HR than the AE group, showing power analysis <80%. At the intragroup level shown in Table 5, there was a substantial increase in HR in the AE-BFR group with power analysis >80%, a substantial reduction in SpO_2_, and an increase in HR after the sessions for both groups.

### 3.2. Symptomatology

Table 6 shows substantial differences between groups after the intervention program, with the AE-BFR group obtaining a greater reduction in stiffness as assessed by the FIQ subscale with power analysis >80%. It also shows substantial differences between groups after the intervention program, with the AE-BFR group obtaining a greater reduction than the AE group in pain assessed by the WPI, Sscore, and FIQ subscale, in fatigue assessed in the dimensions of physical fatigue and reduced activity using the MFI and FIQ subscales of fatigue and morning tiredness, and in global symptomatology assessed by the FIQ total score (all power analysis <80%). At the intragroup level shown in Table 7, there was also a substantial reduction in stiffness in the AE-BFR group as assessed by the FIQ subscale with power analysis >80%. It also shows a substantial reduction in the AE-BFR group after the intervention program in pain assessed by the WPI, Sscore, and FIQ subscale, in fatigue assessed in the dimensions of physical fatigue and reduced activity using the MFI and FIQ subscales of fatigue and morning tiredness, and in global symptomatology assessed by the FIQ total score as well as in work symptoms assessed by the FIQ subscale (all power analysis <80%). In addition, a substantial reduction was observed in anxiety in the AE group, according to the FIQ subscale, and in mental fatigue assessed by the MFI (both power analyses <80%).

### 3.3. Hematological and Biochemical Parameters

Table 8 shows substantial differences between groups after the intervention program, with the AE-BFR group obtaining a greater increase in HDL and Cr and a reduction in CK and LDL levels compared with the AE group, showing all power analyses <80%. The AE group showed a substantial reduction in MDA, CoQ10/Chol, and CoQ10/TG with respect to the AE-BFR group, which showed a power analysis <80%. At the intragroup level shown in Table 9, the AE-BFR group showed a substantial increase in HDL and Cr with power analysis >80%, a substantial increase in CoQ10/LDL, TAC, Chol, and glucose levels, and a substantial reduction in CoQ10/HDL, CK, and LDL levels after the intervention with power analysis <80%. Likewise, the AE group showed a substantial increase in TAC with power analysis >80%. In addition, there was a substantial increase in GOT, HDL, TG, UREA, and glucose levels and a substantial reduction in MDA and uric acid levels, showing all power analyses <80%.

### 3.4. Functional Capacity

Table 10 shows substantial differences between groups after the intervention program in all functional tests, with the AE-BFR group showing a greater increase in balance by TUGT, cardiorespiratory fitness by 6MWT and ISWT, and strength by CST, handgrip, and KES compared with the AE group, showing power analysis >80% in CST, 6MWT, and handgrip. At the intragroup level shown in Table 11, a substantial improvement was observed in the AE-BFR group after the intervention program in all functional capacity tests performed, showing power analysis >80% in TUGT, 6MWT, and CST. However, in the AE group, substantial improvements were only observed after the program in TUGT and CST, showing both power analyses <80%. An adequate power analysis was defined as ≥0.8.

## 4. Discussion

The objective of this pilot study was to evaluate the feasibility, safety, and effects of an intervention based on a low-intensity, low-volume aerobic training program with blood flow restriction with respect to functional capacity and the impact on symptomatology.

The main findings of this study show that the blood flow-restricted aerobic training group for fibromyalgia had a high attendance rate with a low dropout rate, a greater reduction in stiffness, and a greater increase in cardiorespiratory fitness, balance, and strength than the unrestricted aerobic training group, with normal biochemical parameters for both groups. This study was the first to evaluate exercise with blood flow restriction in patients with fibromyalgia. The sample showed similar disease characteristics in terms of symptom severity, autonomy, physical abilities, pharmacotherapy, digestive problems, sleep disorders, emotional state, and, in this case, without hypertension or circulatory problems. All participants met three fundamental minimum criteria in the index of generalized pain and severity of symptoms: symptomatology present at a similar level for no less than 3 months and absence of another pathology that could explain the pain.

The high attendance rate in both groups was similar to that in other studies of aerobic training in patients with fibromyalgia [57]. These attendance and dropout rates may presumably be due to individualized exercise and face-to-face supervision by continuous assessment with the RPE scale and VAS (external focus of attention) [58], which remained in safe ranges for both groups. Both scales showed no exacerbations of acute pain, even though the AE-BFR group developed greater volume over the distance run than the AE group, maintaining SpO_2_ in a safe range [59] and reflecting the absence of hypoxemia. This suggests that support through various scales may be more in line with the intensity, recovery, and self-management needs of patients with fibromyalgia, with greater control, motivation, safety, effects, and feasibility of aerobic training program with blood flow restriction in this population (self-efficacy) [9]. From the perspective of physical exercise and its role in pathophysiology in recent years, there is evidence that physical exercise of moderate-intense intensity and of a certain duration produces negative effects on the intestinal wall, favoring an increase in permeability, and all the harmful effects that this entails: increased local and systemic inflammation, production of proinflammatory cytokines, and alteration of metabolic processes derived from gastrointestinal problems [60]. Likewise, there is also ample evidence that regular aerobic exercise with controlled intensity and duration of exertion has an important modulatory role in improving the composition, diversity, and function of the intestinal microbiota, with a close relationship between the muscle/microbiota gut axis [61]. Regarding this condition, many patients with fibromyalgia present with alterations in the intestinal microbiota due to small intestinal bacterial overgrowth (SIBO) due to an imbalance in the intestinal bacterial flora [62], which in many cases leads to altered intestinal permeability [63,64,65]. This last condition is not new, although it has played a critical role in recent years in the knowledge of pathophysiology because high intestinal permeability is intimately linked to increased low-grade inflammation, metabolic syndrome, autoimmune disorders, neurological alterations, malabsorption, nutritional deficiencies [62,65,66], alterations in diamine oxidase enzyme (DAO) with loss of skeletal muscle strength [67], and increased perceived pain, which has a direct effect on the intestinal microbiota [68]. Therefore, the duration and intensity of exercise in improving fibromyalgia pathophysiology is crucial to avoid counterproductive effects. In this sense, aerobic exercise with low-intensity blood flow restriction maintained at all times the levels of intensity and duration according to the needs of each patient with continuous self-assessment with the VAS perceived pain scale and RPE perceived exertion. In this way, it was possible to control any type of overexertion session by session autonomously.

The cuff pressure exerted in the aerobic training program with blood flow restriction was significantly lower than that used in other populations in various studies [69] because of the tolerability limitations for patients with fibromyalgia assessed by VAS. This confers greater individualization in each session, which adds to the control of perceived exertion and increases the achievement of sufficient stimuli for the improvement of functional capacity in fibromyalgia.

The HR collected in each session reflected that an aerobic training program with blood flow restriction is related to high-intensity training and consequently produces higher VO2 during and after training for a longer time, with the occlusive benefit of safely maintaining HR above the basal range [70]. Another possible reason for the increase in HR may be that people with fibromyalgia present alterations in VO2 kinetics at the muscular level [71], which may result in greater overexertion (reflected in HR), poor VO2max, and a more limited cardiorespiratory condition [72], resulting in a longer recovery after sessions and increased HR.

The improvements in the AE-BFR group compared with the AE group in symptomatologic variables such as pain, fatigue, stiffness, and tiredness agree with other traditional aerobic exercise programs on the impact and symptomatology of fibromyalgia [73,74,75], which corroborated that the intervention period with aerobic training program with blood flow restriction did not exacerbate pain, and perceived exertion was adequately managed in the AE-BFR group. The pathophysiology of fibromyalgia is highly problematic precisely because it encompasses broad and non-specific symptomatology in which pain is the central core. It is a disease in which there is evidence of concentration and memory problems, unrefreshing sleep, muscle and joint stiffness, periods of latency, and exacerbation of symptoms by different types of perceived pain [76], being able to differentiate nociceptive pain, caused mainly by continuous inflammation and tissue damage, neuropathic pain, caused by nerve damage, and nociplastic pain, caused by an alteration in the processing and modulation of pain [77]. In relation to this diverse symptomatology and typology of pain, it is worth highlighting the complexity of the approach and the possible benefit in quality of life that any improvement in the aforementioned aspects entails.

Hematological values remained within normal ranges at the end of the intervention period in both groups. The AE-BFR group showed normal values in those variables most susceptible to excessive training load (CK, Cr, and MDA) [78], and the aerobic training program with blood flow restriction may be safe for patients with fibromyalgia. The AE-BFR group showed improvements in lipid profiles compared with the AE group, presumably because of greater adaptation and stimuli received from more intense aerobic exercise [79] and the blood flow restriction mechanism [80]. Similarly, the AE-BFR group had increased CoQ10/LDL coenzyme ratios compared with the AE group, which may also explain the improved lipid profile due to the antioxidant function of CoQ10 in low-density lipoproteins [81]. The AE group showed a reduction of CoQ10/Chol and CoQ10/TG with respect to the AE-BFR group, which corroborates a worsening of the lipid profile. Therefore, aerobic training programs with blood flow restrictions may be safe and may have a positive impact on cardiovascular health. Likewise, regular low-intensity exercise improves the activity of the cytokine IL-6 and activates the secretion of anti-inflammatory cytokines IL-10, IL-1ra, and TNF-R, thereby improving and regulating metabolic and inflammatory processes and a reduced low-grade inflammatory state [82,83], which in turn also has a positive effect on the cardiovascular system [84]. Therefore, adherence to a training program for patients with fibromyalgia is crucial. The AE-BFR group showed increased TAC, possibly because of an increased level of the coenzyme CoQ10 with respect to the basal state after physical exercise [85]. However, the AE group also showed an increase in TAC, which could be explained by a higher CoQ10 at baseline compared with the AE-BFR group, which could explain their reduction in MDA due to the CoQ10, TAC, and MDA synergy being closely related [86]. Therefore, the aspects discussed on pain, its typology and trigger, the muscle/microbiota axis relationship, proinflammatory cytokines, immunological and hormonal alterations, the importance of the microbiota in the pathophysiology of the disease, and the fundamental role of control of intensity and duration of exercise in the intestinal barrier are currently contextualizing fibromyalgia disease in a neuroimmunoendocrine spectrum [87], where the multifactorial and multidisciplinary role of healthcare professionals is of vital importance in the approach to the disease.

Functional tests showed that the AE-BFR group was superior in all tests of balance, cardiorespiratory fitness, and strength with respect to the AE group, which coincides with studies of aerobic training programs with blood flow restrictions on functional capacity in young people, healthy older adults [88], and pathologies [89]. However, in the present study, the intensity of cuff pressure was assessed by the VAS, and the absolute workload was low, with only 28 min of weekly training. All of this resulted in improvements in functional capacity and greater gait autonomy. This situation was not the same in the AE group, which remained similar in almost all variables, reflecting the fact that these participants did not have sufficient stimuli in volume and intensity to generate sufficient adaptations.

Balance problems, vertigo, and postural instability are frequent in patients with fibromyalgia, presumably due to vestibular, somatosensory, and visual disturbances [90]. Therefore, the performance of the TUGT test is crucial for elucidating the extent of this limitation. In this regard, the reduced test time in the AE-BFR group may be due to an improvement in the CST test, which involves greater strength and endurance of the lower body musculature because of the 9-week intervention period with respect to the AE group. This situation was also observed, although to a lesser extent, in the AE group with reduced time in TUGT and greater cycles completed in CST. Therefore, low-volume, low-intensity aerobic exercise with and without blood flow restriction may be responsible for the reduction of the TUGT test, in line with studies on aerobic exercise in the elderly population and its negative consequences on the balance of sedentary lifestyles [91]. However, in this study, the number of sessions, time per session, and total weeks were less.

As discussed in previous sections, cardiorespiratory fitness in patients with fibromyalgia is limited and negatively affects their daily life and quality of life [4,5,6,11,12]. The 6MWT test is a widespread and reliable method in fibromyalgia to determine the extent of aerobic capacity [41]. In this sense, the AE-BFR group increased the total distance covered with respect to the AE group, presumably because of improvements in the KES test, handgrip, and CST, whose strength values at the end of the intervention period were higher with a greater number of repetitions in the case of CST. This agrees with studies that reflect improvements in aerobic capacity through improvements in strength capacity and vice versa [92]. Likewise, there is related evidence between the strength values reflected in the handgrip test with respect to aerobic capacity [93] and the capacity and muscle mass of the lower body [94]. On the other hand, the ISWT test also showed improvement in the AE-BFR group compared with the AE group. However, in this test, the statistical evidence was lower, presumably because patients with fibromyalgia do not tolerate incremental high-intensity tests correctly [95]. Therefore, blood flow restriction training may be related to the functional improvements produced in cardiorespiratory fitness tests, which is in line with studies where aerobic and neuromuscular improvements have been reported [19,20,23].

The poor neuromuscular condition of patients with fibromyalgia limits their autonomy and quality of life [4,11]. The handgrip, CST, and KES tests show to what extent this condition is limited. The tests showed that the AE-BFR group scored better than the AE group in all tests, with the handgrip and CST being statistically adequate. These improvements in strength scores in the lower and upper body tests agree with studies where a similar situation has been observed [94] and its transfer to aerobic fitness [93]. These improvements suggest that blood flow restriction training may be important in this regard and that as little as 2 days per week can produce such improvements in addition to improved aerobic capacity and balance.

All the functional tests performed after the 9-week intervention period reflected that the training stimulus in the AE-BFR group with the RPE perceived exertion control and the VAS visual pain scale can be beneficial to produce positive changes in the tests without performing any type of intense effort. Effort control, less time spent per session, and rapid post-exertion recovery in patients with fibromyalgia are essential to achieve greater adherence to the exercise program. Therefore, we could say that the results discussed support the initial hypothesis.

### 4.1. Strengths of the Study

The strengths of the study lie in the multidisciplinary team that made up the research with experts in exercise physiology in pathologies that were performed in situ 100% of the intervention period. The medical services of internal medicine and rheumatology, together with the rest of the researchers, adapted the needs of the physical limitations and pain of the patient with fibromyalgia with the necessary adaptations of the blood flow restriction training for its feasibility and safety. Another strong point to highlight is the dropout rate of the AE-BRF group (≤15%) and the relative intensity performed individually in each session with EVA and RPE. This meant knowing in real time the fatigue of the volunteer performing the aerobic training program with blood flow restriction at each moment to facilitate the adjustment of gait intensity and cuff pressure. Another important point is to consider aerobic training with blood flow restriction in women with fibromyalgia as safe, useful, and well-tolerated without adverse effects.

### 4.2. Limitations of the Study

Limitations include a monocentric nature, although with multicentric recruitment of patients, a small representation of samples, and nonrandomization due to contraindications for possible adverse effects based on symptom variability and possible interaction with pharmacological treatment. Another of the strong limitations of the study comes from the power analysis conducted, which reflects a weak significance in several variables, although it showed adequate importance in others, even with a small sample size. These limitations and, in particular, the size of the sample reflect the fact that the study was conducted with an insufficient number of patients; therefore, the data obtained can only be considered at the preliminary level.

Future research is needed to provide more knowledge, with an older sample of younger age and less pharmacological treatment, and that can be random.

## 5. Conclusions

Low-volume, low-intensity blood flow-restricted aerobic training may be feasible, safe, and more effective than unrestricted aerobic training as a tool for physical exercise prescription to improve cardiorespiratory fitness, strength, balance, and stiffness in women with fibromyalgia with common symptomatological and severity characteristics.

## 6. Practical Applications


Use of the visual pain scale (VAS) to individually quantify the cuff pressure of blood flow restriction at each session in patients with fibromyalgia.Use of the rate of perceived exertion (RPE) to quantify the daily effort of blood flow restriction training in patients with fibromyalgia.Prescription of aerobic training with blood flow restriction as an adjuvant tool in fibromyalgia to improve the long-term exercise adherence rate.The prescription of aerobic training with blood flow restriction to improve the lipid profile in pathologies compatible with metabolic syndrome or in people with dyslipidemia.


## Figures and Tables

**Figure 1 biomedicines-12-01895-f001:**
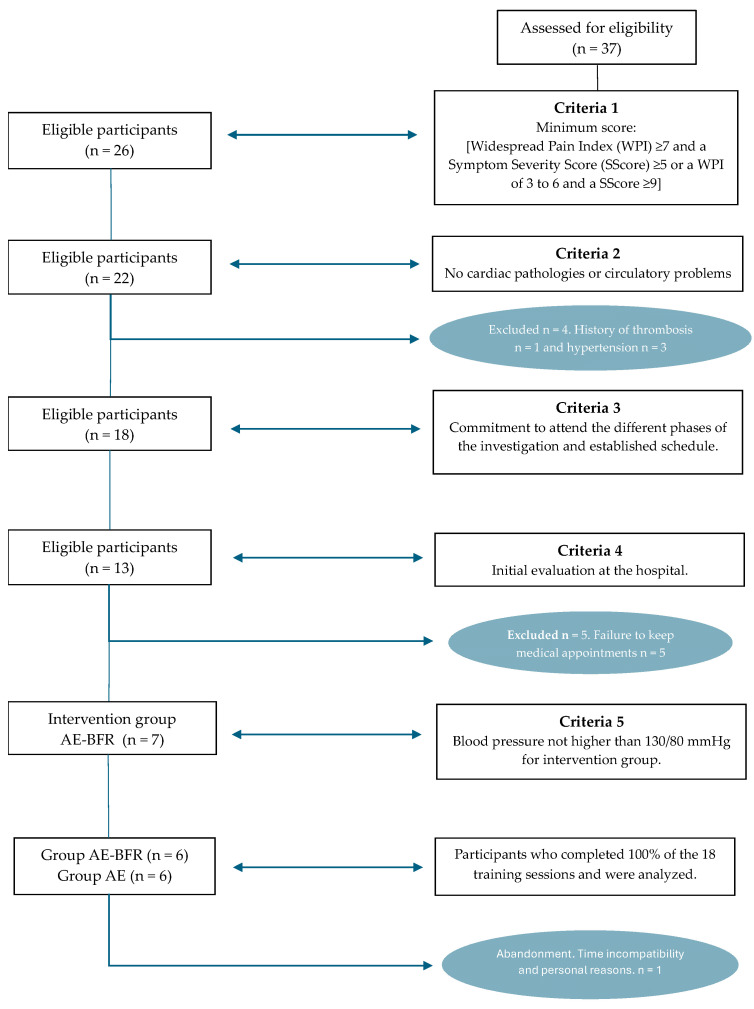
Participants screening.

**Table 1 biomedicines-12-01895-t001:** Characteristics of the participants.

	All (n = 13)	Intervention BFR (n = 7)	Non-Intervention (n = 6)	Compliance BFR (n = 6)	Non-Compliance BFR (n = 1)
Age (years)	59.3 ± 3.2	58.1 ± 2	60.8 ± 3.9	58± 2.1	59 ± 0
Height (cm)	159.7 ± 5.4	161.1 ± 5.1	158.1 ± 5.8	160 ± 5.4	164 ± 0
Mass (kg)	66.5 ± 10.8	66.4 ± 10	66.5 ± 12.5	65.3 ± 10.5	73 ± 0
BMI (kg m^2^)	26 ± 3.9	25.5 ± 3.1	26.5 ± 4.8	25.2 ± 3.4	27.1 ± 0
Systolic BP (mmHg)	128.8 ± 5.8	124.1 ± 2.8	134.3 ± 2.3	124 ± 3	125 ± 0
Diastolic BP (mmHg)	78.4 ± 8	75.5 ± 5.5	83.5 ± 6	74.6 ± 5.4	80 ± 0
* ***Pharmacotherapy***					
Paracetamol 1 g	13 (100)	7 (100)	6 (100)	6 (100)	1 (100)
Pregabalin 75 mg	5 (38.4)	3 (60)	2 (33.3)	2 (33.3)	1 (100)
Ibuprofen 400 mg	13 (100)	7 (100)	6 (100)	6 (100)	1 (100)
Metamizole 575 mg	8 (62)	3 (60)	5 (83.3)	3 (50)	0 (0.0)
Tramadol/paracetamol 37.5 mg + 325 mg	7 (53.8)	3 (42.8)	4 (66.6)	3 (50)	0 (0.0)
Fluoxetine 20 mg	3 (23)	1 (14.2)	2 (33.3)	1 (16.6)	0 (0.0)
Venlafaxina 75 mg	4 (30.7)	1 (14.2)	3 (50)	1 (16.6)	0 (0.0)

Note: Data are presented as mean ± standard deviations or n (%). Abbreviations: BMI, body mass index, BP, blood pressure. Compliance was estimated in the completion of the 18 sessions or 100% of the training period. * Medications previously prescribed by the family medicine, rheumatology, mental health, and internal medicine service during the last 12 months.

**Table 2 biomedicines-12-01895-t002:** Baseline comparisons between characteristics of the participants.

	Group	M (SD)	*p*-Value	Standardized Differences (90% CL)	Qualitative Assessment	Post Hoc Power Analysis
Age (years)	AE-BFR	58.1 (2)	0.177	0.76 (0.97)	85/10/5 *	0.239
	AE	60.8 (3.9)				
Height (cm)	AE-BFR	161.1 (5.1)	0.358	−0.5 (0.94)	10/18/71	0.13
	AE	158.1 (5.8)				
Mass (kg)	AE-BFR	66.4 (10)	0.981	0.01 (0.95)	36/29/35	0.05
	AE	66.5 (12.5)				
BMI (kg m^2^)	AE-BFR	25.5 (3.1)	0.663	0.23 (0.96)	53/26/21	0.67
	AE	26.5 (4.8)				
Systolic BP	AE-BFR	124.1 (2.8)	<0.001	4.93 (0.91)	100/0/0 ***	1 *
	AE	134.3 (2.3)				
Diastolic BP	AE-BFR	75.5 (5.5)	<0.001	3.21 (0.92)	100/0/0 ***	0.999 *
	AE	83.5 (6)				
Paracetamol (1000 mg)	AE-BFR	1000 (0)	1	0 (0)	0/0/0	0.05
	AE	1000 (0)				
Pregabalin (75 mg)	AE-BFR	32 (40)	0.751	−0.17 (0.94)	25/28/48	0.059
	AE	25 (38.7)				
Ibuprofen (400 mg)	AE-BFR	400 (0)	1	0 (0)	0/0/0	0.05
	AE	400 (0)				
Metamizole (575 mg)	AE-BFR	246.4 (307)	0.153	0.79 (0.93)	85/10/4	0.254
	AE	479.1 (234.7)				
Tramadol/paracetamol (37.5 mg/325 mg)	AE-BFR	135.3 (193.7)	0.434	0.42 (0.94)	66/21/13	0.106
	AE	242.1 (187.5)				
Fluoxetine 20 (mg)	AE-BFR	2.8 (7.5)	0.474	0.39 (0.95)	64/22/14	0.098
	AE	6.6 (10.3)				
Venlafaxina (75 mg)	AE-BFR	10.7 (28.3)	0.215	0.7 (0.96)	82/12/6 *	0.21
	AE	37.5 (41)				

Abbreviations: Confidence level (CL), mean (M), standard deviation (SD), Aerobic exercise with blood flow restriction group (AE-BFR), Aerobic exercise group (AE), body mass index (BMI), blood pressure (BP). Asterisks indicate significant for baseline comparisons when *p* < 0.002 (0.05/25 comparisons = 0.002. The numbers of asterisks indicate the likelihood for the baseline characteristics of the participants between-group differences to be substantial, with 1 symbol (*) referring to likely differences, and 3 symbols (***) to almost certain differences. Threshold values for Cohen’s ES were trivial (0.0–0.19), small (0.20–0.59), moderate (0.60–1.19), large (1.20–1.99), and very large (>2.0). Numbers shown are the quantitative chances (%) that the effect is positive/trivial/negative. A substantial effect was defined as >75%. An adequate power analysis was defined as ≥0.8.

**Table 3 biomedicines-12-01895-t003:** Comparisons between aerobic exercise with blood flow restriction and aerobic exercise groups in distance and RPE measures.

	Group	M (SD)	*p*-Value	Standardized Differences (90% CL)	Qualitative Assessment	Post Hoc Power Analysis
Distance, meters	AE-BFR	911 (303)	<0.001	−0.77 (0.12)	0/0/100 ***	0.334
	AE	675 (180)				
RPE, 0–10	AE-BFR	6.03 (1.36)	0.117	−0.24 (0.25)	0/40/60	0.104
	AE	5.70 (1.60)				

Abbreviations: Confidence level (CL), mean (M), standard deviation (SD), Aerobic exercise with blood flow restriction group (AE-BFR), Aerobic exercise group (AE), Rate of Perceived Exertion (RPE). The distances in meters are those relative to the 18 sessions of the intervention period, together with the RPE. Asterisks indicate significant for between-group comparisons when *p* < 0.002 (0.05/25 comparisons = 0.002). The numbers of asterisks indicate the likelihood for the between-group differences to be substantial, with 3 symbols (***) to almost certain differences. Threshold values for Cohen’s ES were trivial (0.0–0.19), small (0.20–0.59), moderate (0.60–1.19), large (1.20–1.99), and very large (>2.0). Numbers shown are the quantitative chances (%) that the effect is positive/trivial/negative. A substantial effect was defined as >75%. An adequate power analysis was defined as ≥0.8.

**Table 4 biomedicines-12-01895-t004:** Changes between groups in SPO_2_, HR, and VAS parameters produced in response to the training intervention.

	Group	PRE M (SD)	POST M (SD)	*p* (Group × Time)	Between-Group Standardized Differences (90% CL)	Between-Group Qualitative Assessment of Difference	Post Hoc Power Analysis
SpO_2_	AE-BFR	98.64 (0.63)	98.09 (1.13)	0.255	0.17 (0.25)	43/56/1	0.085
	AE	97.97 (1.08)	97.64 (1.34)				
HR	AE-BFR	77.6 (10.5)	98.7 (12.4)	<0.001	−0.96 (0.19)	0/0/100 ***	0.462
	AE	72.7 (11.9)	82.8 (13.6)				
VAS	AE-BFR	3.60 (1.84)	3.72 (1.76)	0.999	0.00 (0.23)	8/85/7	0.050
	AE	3.99 (1.46)	4.11 (1.47)				

Abbreviations: Confidence level (CL), mean (M), standard deviation (SD), Aerobic exercise with blood flow restriction group (AE-BFR), Aerobic exercise group (AE), oxygen saturation (SpO_2_), heart rate (HR), Visual Analog Scale (VAS). Asterisks indicate significant for between-group comparisons when *p* < 0.002 (0.05/25 comparisons = 0.002. The numbers of asterisks indicate the likelihood for the between-group differences to be substantial, with 3 symbols (***) to almost certain differences. Threshold values for Cohen’s ES were trivial (0.0–0.19), small (0.20–0.59), moderate (0.60–1.19), large (1.20–1.99), and very large (>2.0). Numbers shown are the quantitative chances (%) that the effect is positive/trivial/negative. A substantial effect was defined as >75%. An adequate power analysis was defined as ≥0.8.

**Table 5 biomedicines-12-01895-t005:** Intragroup changes in SpO_2_, HR, and VAS parameters produced in response to the training intervention.

	Group	PRE M (SD)	POST M (SD)	*p* Pre-Post (Intra-Group)	Standardized Differences (90% CL)	Qualitative Assessment	Post Hoc Power Analysis
SpO_2_	AE-BFR	98.64 (0.63)	98.09 (1.13)	<0.001	−0.86 (0.30)	0/0/100 ***	0.399
	AE	97.97 (1.08)	97.64 (1.34)	0.035	−0.31 (0.24)	0/23/77 *	0.126
HR	AE-BFR	77.6 (10.5)	98.7 (12.4)	<0.001	2.00 (0.17)	100/0/0 ***	0.942 *
	AE	72.7 (11.9)	82.8 (13.6)	<0.001	0.84 (0.14)	100/0/0 ***	0.386
VAS	AE-BFR	3.60 (1.84)	3.72 (1.76)	0.272	0.06 (0.10)	1/99/0 **	0.061
	AE	3.99 (1.46)	4.11 (1.47)	0.289	0.08 (0.13)	6/94/0 *	0.065

Abbreviations: Confidence level (CL), mean (M), standard deviation (SD), Aerobic exercise with blood flow restriction group (AE-BFR), Aerobic exercise group (AE), oxygen saturation (SpO_2_), heart rate (HR), Visual Analog Scale (VAS). Asterisks indicate significant for intragroup comparisons when *p* < 0.002 (0.05/25 comparisons = 0.002. The numbers of asterisks indicate the likelihood for the intragroup differences to be substantial, with 1 symbol (*) referring to likely differences, 2 symbols (**) to very likely, and 3 symbols (***) to almost certain differences. Threshold values for Cohen’s ES were trivial (0.0–0.19), small (0.20–0.59), moderate (0.60–1.19), large (1.20–1.99), and very large (>2.0). Numbers shown are the quantitative chances (%) that the effect is positive/trivial/negative. A substantial effect was defined as >75%. SpO_2_, HR, and VAS are those related to the 18 sessions of the intervention period. An adequate power analysis was defined as ≥0.8.

**Table 6 biomedicines-12-01895-t006:** Changes between groups in symptomatology produced in response to the training intervention.

	Group	PRE M (SD)	POST M (SD)	*p* (Group × Time)	Between-Group Standardized Differences (90% CL)	Between-Group Qualitative Assessment of Difference	Post Hoc Power Analysis
WPI	AE-BFR	12.3 (3.7)	8.2 (4.2)	0.045	1.25 (0.97)	96/3/1 **	0.645
	AE	12.3 (3.9)	12.2 (4.8)				
Sscore	AE-BFR	7.8 (2.3)	5.3 (2.7)	0.240	0.66 (0.98)	80/13/7 *	0.281
	AE	6.5 (1.9)	6.2 (1.9)				
FIQ total	AE-BFR	48.3 (22.2)	32.2 (13.3)	0.091	1.01 (0.98)	92/5/3 *	0.494
	AE	47.7 (20.9)	48.0 (19.0)				
FIQ feel good	AE-BFR	4.76 (3.58)	3.34 (1.95)	0.383	0.48 (0.97)	70/19/11	0.192
	AE	3.30 (3.19)	3.29 (2.13)				
FIQ physical function	AE-BFR	3.48 (2.32)	3.14 (1.94)	0.750	0.17 (0.98)	48/27/25	0.085
	AE	4.29 (1.84)	4.27 (2.03)				
FIQ work	AE-BFR	4.35 (2.69)	2.75 (1.73)	0.911	0.06 (0.98)	40/28/32	0.061
	AE	5.97 (2.48)	4.57 (2.92)				
FIQ pain	AE-BFR	4.83 (2.00)	3.63 (1.84)	0.430	0.44 (0.97)	67/20/13	0.175
	AE	6.10 (2.30)	5.75 (1.96)				
FIQ fatigue	AE-BFR	5.95 (2.03)	4.47 (1.94)	0.166	0.81 (1.00)	86/9/5 *	0.368
	AE	6.73 (1.88)	7.20 (2.29)				
FIQ morning tiredness	AE-BFR	5.97 (1.84)	3.45 (2.21)	0.027	1.39 (0.97)	97/2/1 **	0.724
	AE	5.72 (2.48)	6.58 (2.64)				
FIQ stiffness	AE-BFR	5.80 (1.25)	2.12 (1.12)	0.002	2.39 (0.98)	100/0/0 ***	0.986 *
	AE	5.68 (2.20)	6.20 (2.07)				
FIQ anxiety	AE-BFR	5.32 (2.62)	4.32 (2.11)	0.893	−0.07 (0.98)	30/29/40	0.063
	AE	4.87 (2.80)	3.68 (2.96)				
FIQ depression	AE-BFR	4.52 (2.23)	3.28 (1.80)	0.833	0.11 (0.99)	44/28/28	0.071
	AE	3.95 (2.93)	3.00 (2.08)				
MFI general fatigue	AE-BFR	14.3 (1.4)	12.2 (2.1)	0.365	0.51 (0.98)	71/18/11	0.206
	AE	16.0 (1.9)	15.0 (3.2)				
MFI physical fatigue	AE-BFR	13.8 (2.1)	12.3 (2.8)	0.254	0.63 (0.98)	79/14/7 *	0.265
	AE	12.8 (2.5)	13.0 (1.8)				
MFI reduced activity	AE-BFR	10.8 (3.9)	9.3 (2.4)	0.177	0.79 (1.01)	85/9/5 *	0.356
	AE	9.7 (3.1)	10.0 (3.1)				
MFI reduced motivation	AE-BFR	13.2 (3.2)	12.3 (2.3)	0.903	−0.07 (0.97)	31/28/40	0.063
	AE	10.7 (2.1)	9.7 (3.5)				
MFI mental fatigue	AE-BFR	15.8 (1.9)	15.3 (2.3)	0.463	−0.41 (0.97)	14/21/65	0.163
	AE	14.5 (1.9)	13.0 (2.6)				

Abbreviations: Confidence level (CL), mean (M), standard deviation (SD), Aerobic exercise with blood flow restriction group (AE-BFR), Aerobic exercise group (AE), Widespread Pain Index (WPI), Symptom Severity Score (Sscore), Fibromyalgia Impact Questionnaire (FIQ), Multidimensional Fatigue Inventory (MFI). Asterisks indicate significant for between-group comparisons when *p* < 0.002 (0.05/25 comparisons = 0.002. The numbers of asterisks indicate the likelihood for the between-group differences to be substantial, with 1 symbol (*) referring to likely differences, 2 symbols (**) to very likely, and 3 symbols (***) to almost certain differences. Threshold values for Cohen’s ES were trivial (0.0–0.19), small (0.20–0.59), moderate (0.60–1.19), large (1.20–1.99), and very large (>2.0). Numbers shown are the quantitative chances (%) that the effect is positive/trivial/negative. A substantial effect was defined as >75%. An adequate power analysis was defined as ≥0.8.

**Table 7 biomedicines-12-01895-t007:** Intragroup changes in symptomatology produced in response to the training intervention.

	Group	PRE M (SD)	POST M (SD)	*p* Pre-Post (Intra-Group)	Standardized Differences (90% CL)	Qualitative Assessment	Post Hoc Power Analysis
WPI	AE-BFR	12.3 (3.7)	8.2 (4.2)	0.038	−0.96 (0.69)	1/3/96 **	0.462
	AE	12.3 (3.9)	12.2 (4.8)	0.822	−0.04 (0.30)	9/75/16	0.057
Sscore	AE-BFR	7.8 (2.3)	5.3 (2.7)	0.171	−0.91 (1.15)	5/8/87 *	0.430
	AE	6.5 (1.9)	6.2 (1.9)	0.576	−0.15 (0.51)	11/46/43	0.080
FIQ total	AE-BFR	48.3 (22.2)	32.2 (13.3)	0.084	−0.61 (0.57)	2/9/90 *	0.255
	AE	47.7 (20.9)	48.0 (19.0)	0.941	0.01 (0.31)	14/75/11	0.052
FIQ physical function	AE-BFR	3.48 (2.32)	3.14 (1.94)	0.716	−0.12 (0.64)	18/41/41	0.073
	AE	4.29 (1.84)	4.27 (2.03)	0.973	−0.01 (0.39)	17/65/18	0.052
FIQ feel good	AE-BFR	4.76 (3.58)	3.34 (1.95)	0.350	−0.34 (0.65)	8/27/65	0.137
	AE	3.30 (3.19)	3.29 (2.13)	0.988	0.00 (0.34)	14/71/15	0.050
FIQ work	AE-BFR	4.35 (2.69)	2.75 (1.73)	0.240	−0.50 (0.76)	6/17/77 *	0.201
	AE	5.97 (2.48)	4.57 (2.92)	0.316	−0.48 (0.86)	9/19/73	0.192
FIQ pain	AE-BFR	4.83 (2.00)	3.63 (1.84)	0.196	−0.50 (0.68)	5/16/80 *	0.201
	AE	6.10 (2.30)	5.75 (1.96)	0.609	−0.13 (0.47)	11/50/39	0.076
FIQ fatigue	AE-BFR	5.95 (2.03)	4.47 (1.94)	0.246	−0.62 (0.95)	7/14/79 *	0.260
	AE	6.73 (1.88)	7.20 (2.29)	0.400	0.21 (0.46)	52/42/7	0.096
FIQ morning tiredness	AE-BFR	5.97 (1.84)	3.45 (2.21)	0.020	−1.15 (0.69)	1/1/98 **	0.583
	AE	5.72 (2.48)	6.58 (2.64)	0.444	0.29 (0.71)	60/29/11	0.120
FIQ stiffness	AE-BFR	5.80 (1.25)	2.12 (1.12)	0.003	−2.49 (0.92)	0/0/100 ***	0.991 *
	AE	5.68 (2.20)	6.20 (2.07)	0.462	0.20 (0.50)	50/42/9	0.093
FIQ anxiety	AE-BFR	5.32 (2.62)	4.32 (2.11)	0.455	−0.32 (0.80)	12/26/61	0.130
	AE	4.87 (2.80)	3.68 (2.96)	0.035	−0.36 (0.25)	0/13/87 *	0.144
FIQ depression	AE-BFR	4.52 (2.23)	3.28 (1.80)	0.323	−0.47 (0.86)	9/19/72	0.188
	AE	3.95 (2.93)	3.00 (2.08)	0.198	−0.27 (0.37)	3/33/65	0.114
MFI general fatigue	AE-BFR	14.3 (1.4)	12.2 (2.1)	0.056	−1.34 (1.08)	2/3/96 **	0.697
	AE	16.0 (1.9)	15.0 (3.2)	0.296	−0.44 (0.77)	8/20/73	0.175
MFI physical fatigue	AE-BFR	13.8 (2.1)	12.3 (2.8)	0.287	−0.59 (1.00)	9/15/77 *	0.245
	AE	12.8 (2.5)	13.0 (1.8)	0.695	0.06 (0.27)	17/77/6	0.061
MFI reduced activity	AE-BFR	10.8 (3.9)	9.3 (2.4)	0.237	−0.33 (0.49)	4/27/69	0.133
	AE	9.7 (3.1)	10.0 (3.1)	0.363	0.09 (0.18)	14/84/1	0.067
MFI reduced motivation	AE-BFR	13.2 (3.2)	12.3 (2.3)	0.449	−0.22 (0.54)	9/38/53	0.099
	AE	10.7 (2.1)	9.7 (3.5)	0.300	−0.41 (0.70)	7/22/71	0.163
MFI mental fatigue	AE-BFR	15.8 (1.9)	15.3 (2.3)	0.542	−0.22 (0.67)	13/35/52	0.099
	AE	14.5 (1.9)	13.0 (2.6)	0.215	−0.68 (0.96)	6/12/82 *	0.292

Abbreviations: Confidence level (CL), mean (M), standard deviation (SD), Aerobic exercise with blood flow restriction group (AE-BFR), Aerobic exercise group (AE), Widespread Pain Index (WPI), Symptom Severity Score (Sscore), Fibromyalgia Impact Questionnaire (FIQ), Multidimensional Fatigue Inventory (MFI). Asterisks indicate significant for intragroup comparisons when *p* < 0.002 (0.05/25 comparisons = 0.002. The numbers of asterisks indicate the likelihood for the intragroup differences to be substantial, with 1 symbol (*) referring to likely differences, 2 symbols (**) to very likely, and 3 symbols (***) to almost certain differences. Threshold values for Cohen’s ES were trivial (0.0–0.19), small (0.20–0.59), moderate (0.60–1.19), large (1.20–1.99), and very large (>2.0). Numbers shown are the quantitative chances (%) that the effect is positive/trivial/negative. A substantial effect was defined as >75%. An adequate power analysis was defined as ≥0.8.

**Table 8 biomedicines-12-01895-t008:** Changes between groups in hematological and biochemical parameters produced in response to the training intervention.

	Group	PRE M (SD)	POST M (SD)	*p* (Group × Time)	Between-Group Standardized Differences (90% CL)	Between-Group Qualitative Assessment of Difference	Post Hoc Power Analysis
MDA µmol/L	AE-BFR	2.37 (0.59)	2.46 (0.40)	0.217	−0.71 (0.98)	6/12/82 *	0.309
	AE	3.38 (1.09)	2.77 (0.33)				
CK ukat/L	AE-BFR	0.80 (0.20)	0.59 (0.23)	0.297	0.58 (0.97)	76/16/9 *	0.240
	AE	0.58 (0.18)	0.58 (0.17)				
GGT ukat/L	AE-BFR	0.21 (0.07)	0.23 (0.09)	0.593	0.28 (1.00)	56/25/19	0.117
	AE	0.43 (0.22)	0.58 (0.73)				
GPT ukat/L	AE-BFR	0.14 (0.07)	0.17 (0.10)	0.845	−0.11 (0.97)	29/28/43	0.071
	AE	0.19 (0.12)	0.19 (0.04)				
GOT ukat/L	AE-BFR	0.26 (0.10)	0.26 (0.13)	0.628	0.26 (0.99)	55/25/20	0.110
	AE	0.22 (0.04)	0.25 (0.08)				
ALP ukat/L	AE-BFR	0.89 (0.16)	1.08 (0.52)	0.396	−0.47 (0.98)	12/19/69	0.188
	AE	1.07 (0.44)	1.01 (0.35)				
Chol mmol/L	AE-BFR	4.33 (0.75)	5.04 (1.17)	0.703	−0.21 (0.97)	23/26/51	0.096
	AE	4.42 (1.06)	4.84 (0.52)				
HDL mmol/L	AE-BFR	1.45 (0.36)	2.19 (0.31)	0.070	−1.09 (0.97)	2/4/94 *	0.545
	AE	1.64 (0.31)	1.84 (0.15)				
TG mmol/L	AE-BFR	1.40 (0.69)	1.71 (0.36)	0.888	0.08 (0.98)	41/28/31	0.065
	AE	1.27 (0.79)	1.67 (0.80)				
LDL mmol/L	AE-BFR	2.73 (0.73)	2.36 (0.72)	0.258	0.63 (0.98)	79/14/7 *	0.265
	AE	2.54 (0.72)	2.67 (0.34)				
Urea mmol/L	AE-BFR	3.41 (1.14)	3.71 (2.09)	0.091	1.01 (0.98)	92/6/3 *	0.494
	AE	3.84 (2.57)	6.64 (2.37)				
Cr umol/L	AE-BFR	74.27 (9.67)	92.84 (8.71)	0.177	−0.77 (0.97)	5/10/85 *	0.344
	AE	83.11 (12.6)	89.3 (10.57)				
Glucose mmol/L	AE-BFR	4.02 (0.34)	4.48 (0.56)	0.328	0.55 (0.97)	74/17/10	0.225
	AE	4.23 (0.96)	5.18 (1.10)				
Uric ac mmol/L	AE-BFR	0.22 (0.09)	0.18 (0.04)	0.433	−0.44 (0.97)	13/20/67	0.175
	AE	0.28 (0.06)	0.20 (0.04)				
CoQ10 nmol/L	AE-BFR	444.7 (149.9)	499.8 (91.0)	0.312	−0.56 (0.98)	9/16/75	0.230
	AE	736.1 (678.6)	611.0 (319.2)				
CoQ_10_/Chol, nmol/mmol	AE-BFR	92.07 (23.71)	92.19 (22.17)	0.379	−0.90 (1.88)	15/10/76 *	0.424
	AE	149.47 (119.25)	118.10 (70.75)				
CoQ_10_/HDL, nmol/mmol	AE-BFR	295.57 (158.44)	207.06 (31.54)	0.936	−0.05 (1.29)	35/23/41	0.059
	AE	401.16 (329.57)	302.26 (159.75)				
CoQ_10_/LDL, nmol/mmol	AE-BFR	150.7 (46.19)	203.31 (65.55)	0.407	−0.56 (1.24)	14/16/71	0.230
	AE	258.74 (194.60)	215.93 (135.12)				
CoQ_10_/TG, nmol/mmol	AE-BFR	350.49 (154.47)	339.77 (147.73)	0.119	−1.48 (1.58)	4/4/92 *	0.770
	AE	578.93 (443.81)	396.19 (162.67)				
TAC mmol/L	AE-BFR	0.16 (0.08)	0.25 (0.03)	0.547	0.33 (0.98)	59/23/17	0.133
	AE	0.17 (0.05)	0.29 (0.03)				

Abbreviations: Confidence level (CL), mean (M), standard deviation (SD), Aerobic exercise with blood flow restriction group (AE-BFR), Aerobic exercise group (AE), malondialdehyde (MDA), creatine kinase (CK), gamma-glutamyl transferase (GGT), glutamate-pyruvate transaminase (GPT), glutamic-oxaloacetic transaminase (GOT), alkaline phosphatase (ALP), total cholesterol (Chol), high-density lipoproteins (HDL), triglycerides (TG), low-density lipoprotein (LDL), creatinine (Cr), Uric acid (Uric ac), coenzyme Q-10 total (CoQ10), total antioxidant capacity (TAC) in reference to mM TROLOX. Asterisks indicate significant for between-group comparisons when *p* < 0.002 (0.05/25 comparisons = 0.002. The numbers of asterisks indicate the likelihood for the between-group differences to be substantial, with 1 symbol (*) referring to likely differences. Threshold values for Cohen’s ES were trivial (0.0–0.19), small (0.20–0.59), moderate (0.60–1.19), large (1.20–1.99), and very large (>2.0). Numbers shown are the quantitative chances (%) that the effect is positive/trivial/negative. A substantial effect was defined as >75%. An adequate power analysis was defined as ≥0.8.

**Table 9 biomedicines-12-01895-t009:** Intragroup changes in blood hematological and biochemical parameters produced in response to the training intervention.

	Group	PRE M (SD)	POST M (SD)	*p* Pre-Post (Intra-Group)	Standardized Differences (90% CL)	Qualitative Assessment	Post Hoc Power Analysis
CoQ_10_ nmol/L	AE-BFR	444.7 (149.9)	499.8 (91.0)	0.405	0.31 (0.69)	62/28/10	0.126
	AE	736.1 (678.6)	611.0 (319.2)	0.447	−0.16 (0.38)	6/53/41	0.083
CoQ_10_/Chol nmol/mmol	AE-BFR	92.07 (23.71)	92.19 (22.17)	0.992	0.00 (0.86)	33/34/33	0.050
	AE	149.47 (119.25)	118.10 (70.75)	0.245	−0.22 (0.34)	3/42/55	0.099
CoQ_10_/HDL nmol/mmol	AE-BFR	295.5 (158.44)	207.06 (31.54)	0.237	−0.47 (0.71)	6/18/76 *	0.188
	AE	401.16 (329.57)	302.26 (159.75)	0.251	−0.25 (0.39)	3/36/60	0.107
CoQ_10_/TG nmol/mmol	AE-BFR	350.49 (154.47)	339.77 (147.73)	0.924	−0.06 (1.17)	34/25/41	0.061
	AE	578.93 (443.81)	396.19 (162.67)	0.274	−0.35 (0.57)	6/26/69	0.140
CoQ_10_/LDL nmol/mmol	AE-BFR	150.70 (46.19)	203.31 (65.55)	0.064	0.96 (0.82)	94/4/2 *	0.462
	AE	258.74 (194.60)	215.93 (135.12)	0.272	−0.19 (0.30)	3/51/46	0.091
TAC mmol/L	AE-BFR	0.16 (0.08)	0.25 (0.03)	0.084	0.96 (0.90)	93/5/2 *	0.462
	AE	0.17 (0.05)	0.29 (0.03)	0.003	1.85 (0.70)	100/0/0 ***	0.909 *
MDA µmol/L	AE-BFR	2.37 (0.59)	2.46 (0.40)	0.808	0.13 (1.01)	45/28/27	0.076
	AE	3.38 (1.09)	2.77 (0.33)	0.184	−0.47 (0.62)	4/17/79 *	0.188
CK ukat/L	AE-BFR	0.80 (0.20)	0.59 (0.23)	0.252	−0.90 (1.41)	9/9/82 *	0.424
	AE	0.58 (0.18)	0.58 (0.17)	0.998	0.00 (0.92)	34/32/34	0.050
GGT ukat/L	AE-BFR	0.21 (0.07)	0.23 (0.09)	0.717	0.23 (1.23)	52/22/25	0.102
	AE	0.43 (0.22)	0.58 (0.73)	0.531	0.60 (1.79)	66/13/21	0.250
GPT ukat/L	AE-BFR	0.14 (0.07)	0.17 (0.10)	0.759	0.24 (1.50)	52/19/29	0.104
	AE	0.19 (0.12)	0.19 (0.04)	0.937	0.03 (0.72)	33/40/28	0.055
GOT ukat/L	AE-BFR	0.26 (0.10)	0.26 (0.13)	0.945	−0.05 (1.37)	36/22/42	0.059
	AE	0.22 (0.04)	0.25 (0.08)	0.303	−0.81 (1.42)	79/11/11 *	0.368
ALP ukat/L	AE-BFR	0.89 (0.16)	1.08 (0.52)	0.468	0.98 (2.51)	72/9/19	0.475
	AE	1.07 (0.44)	1.01 (0.35)	0.673	−0.11 (0.47)	12/52/35	0.071
Chol mmol/L	AE-BFR	4.33 (0.75)	5.04 (1.17)	0.186	0.80 (1.05)	85/9/6 *	0.362
	AE	4.42 (1.06)	4.84 (0.52)	0.482	0.34 (0.89)	62/24/14	0.137
HDL mmol/L	AE-BFR	1.45 (0.36)	2.19 (0.31)	0.014	1.71 (0.93)	99/1/0 **	0.866 *
	AE	1.64 (0.31)	1.84 (0.15)	0.283	0.54 (0.91)	76/16/8 *	0.220
TG mmol/L	AE-BFR	1.40 (0.69)	1.71 (0.36)	0.540	0.38 (1.17)	62/20/18	0.151
	AE	1.27 (0.79)	1.67 (0.80)	0.372	0.93 (1.92)	76/9/14 *	0.430
LDL mmol/L	AE-BFR	2.73 (0.73)	2.36 (0.72)	0.089	−0.43 (0.41)	1/14/85 *	0.171
	AE	2.54 (0.72)	2.67 (0.34)	0.735	0.16 (0.87)	46/31/22	0.083
Urea mmol/L	AE-BFR	3.41 (1.14)	3.71 (2.09)	0.796	0.22 (1.62)	51/18/31	0.099
	AE	3.84 (2.57)	6.64 (2.37)	0.011	0.92 (0.47)	99/1/0 **	0.436
Cr umol/L	AE-BFR	74.27 (9.67)	92.84 (8.71)	0.012	1.63 (0.85)	99/1/0 **	0.836 *
	AE	83.11 (12.6)	89.3 (10.57)	0.379	0.44 (0.91)	69/20/11	0.175
Glucose mmol/L	AE-BFR	4.02 (0.34)	4.48 (0.56)	0.159	1.15 (1.40)	88/6/6 *	0.583
	AE	4.23 (0.96)	5.18 (1.10)	0.054	0.83 (0.67)	94/4/1 *	0.380
Uric ac mmol/L	AE-BFR	0.22 (0.09)	0.18 (0.04)	0.329	−0.38 (0.71)	8/23/69	0.151
	AE	0.28 (0.06)	0.20 (0.04)	0.050	−1.03 (0.81)	1/3/95 **	0.507

Abbreviations: Confidence level (CL), mean (M), standard deviation (SD), Aerobic exercise with blood flow restriction group (AE-BFR), Aerobic exercise group (AE), malondialdehyde (MDA), creatine kinase (CK), gamma-glutamyl transferase (GGT), glutamate-pyruvate transaminase (GPT), glutamic-oxaloacetic transaminase (GOT), alkaline phosphatase (ALP), total cholesterol (Chol), high-density lipoproteins (HDL), triglycerides (TG), low-density lipoprotein (LDL), creatinine (Cr), Uric acid (Uric ac), coenzyme Q-10 total (CoQ10), total antioxidant capacity (TAC) in reference to mM TROLOX. Asterisks indicate significant for intragroup comparisons when *p* < 0.002 (0.05/25 comparisons = 0.002. The numbers of asterisks indicate the likelihood for the intragroup differences to be substantial, with 1 symbol (*) referring to likely differences, 2 symbols (**) to very likely, and 3 symbols (***) to almost certain differences. Threshold values for Cohen’s ES were trivial (0.0–0.19), small (0.20–0.59), moderate (0.60–1.19), large (1.20–1.99), and very large (>2.0). Numbers shown are the quantitative chances (%) that the effect is positive/trivial/negative. A substantial effect was defined as >75%. An adequate power analysis was defined as ≥0.8.

**Table 10 biomedicines-12-01895-t010:** Changes between groups in functional capacity produced in response to the training intervention.

	Group	PRE M (SD)	POST M (SD)	*p* (Group × Time)	Between-Group Standardized Differences (90% CL)	Between-Group Qualitative Assessment of Difference	Post Hoc Power Analysis
TUGT (s)	AE-BFR	7.72 (0.78)	5.20 (0.91)	0.074	1.08 (0.98)	93/5/2 *	0.539
	AE	8.52 (1.10)	7.55 (1.08)				
CST (reps)	AE-BFR	11.2 (1.2)	17.7 (2.9)	0.005	−2.00 (0.97)	0/0/100 ***	0.942 *
	AE	12.2 (1.8))	13.5 (2.2)				
6MWT (meters)	AE-BFR	423.4 (33.7)	513.0 (31.8)	0.011	−1.69 (0.97)	0/1/99 **	0.859 *
	AE	401.7 (51.5)	419.4 (59.5)				
ISWT (meters)	AE-BFR	568.6 (112.7)	669.4 (47.3)	0.182	−0.76 (1.00)	5/10/85 *	0.338
	AE	389.3 (121.6)	393.9 (116.3)				
Handgrip/BW (Kg)	AE-BFR	0.36 (0.06)	0.45 (0.07)	0.012	−1.67 (0.97)	0/1/99 **	0.852 *
	AE	0.32 (0.07)	0.29 (0.07)				
KES/BW (N)	AE-BFR	2.65 (1.20)	3.43 (0.63)	0.284	−0.60 (0.98)	8/15/77 *	0.250
	AE	2.28 (1.12)	2.34 (0.98)				

Abbreviations: Confidence level (CL), mean (M), standard deviation (SD), Aerobic exercise with blood flow restriction group (AE-BFR), Aerobic exercise group (AE), newton (N), timed up-and-go test (TUGT), 30 s chair stand test (CST), 6 min walk test (6MWT), incremental shuttle walk test (ISWT), body weight (BW), knee extension strength (KES). Asterisks indicate significant for between-group comparisons when *p* < 0.002 (0.05/25 comparisons = 0.002. The numbers of asterisks indicate the likelihood for the between-group differences to be substantial, with 1 symbol (*) referring to likely differences, 2 symbols (**) to very likely, and 3 symbols (***) to almost certain differences. Threshold values for Cohen’s ES were trivial (0.0–0.19), small (0.20–0.59), moderate (0.60–1.19), large (1.20–1.99), and very large (>2.0). Numbers shown are the quantitative chances (%) that the effect is positive/trivial/negative. A substantial effect was defined as >75%. An adequate power analysis was defined as ≥0.8.

**Table 11 biomedicines-12-01895-t011:** Intragroup changes in functional capacity produced in response to the training intervention.

	Group	PRE M (SD)	POST M (SD)	*p* Pre-Post (Intra-Group)	Standardized Differences (90% CL)	Qualitative Assessment	Post Hoc Power Analysis
TUGT (s)	AE-BFR	7.72 (0.78)	5.20 (0.91)	0.006	−2.71 (1.18)	0/0/100 ***	0.997 *
	AE	8.52 (1.10)	7.55 (1.08)	0.135	−0.74 (0.84)	4/9/87 *	0.326
CST (reps)	AE-BFR	11.2 (1.2)	17.7 (2.9)	0.002	4.68 (1.53)	100/0/0 ***	1.000 *
	AE	12.2 (1.8))	13.5 (2.2)	0.191	0.61 (0.82)	82/13/5 *	0.255
6MWT (meters)	AE-BFR	423.4 (33.7)	513.0 (31.8)	0.004	2.24 (0.92)	100/0/0 ***	0.975 *
	AE	401.7 (51.5)	419.4 (59.5)	0.234	0.29 (0.43)	65/31/4	0.120
ISWT (meters)	AE-BFR	568.6 (112.7)	669.4 (47.3)	0.158	0.75 (0.92)	86/9/5 *	0.332
	AE	389.3 (121.6)	393.9 (116.3)	0.736	0.03 (0.18)	6/92/2	0.055
Handgrip/BW (Kg)	AE-BFR	0.36 (0.06)	0.45 (0.07)	0.016	1.25 (0.71)	98/1/0 **	0.645
	AE	0.32 (0.07)	0.29 (0.07)	0.294	−0.43 (0.74)	7/21/72	0.171
KES/BW (N)	AE-BFR	2.65 (1.20)	3.43 (0.63)	0.222	0.55 (0.79)	79/15/6 *	0.225
	AE	2.28 (1.12)	2.34 (0.98)	0.844	0.04 (0.42)	24/61/15	0.057

Abbreviations: Confidence level (CL), mean (M), standard deviation (SD), Aerobic exercise with blood flow restriction group (AE-BFR), Aerobic exercise group (AE), newton (N), timed up-and-go test (TUGT), 30 s chair stand test (CST), 6 min walk test (6MWT), incremental shuttle walk test (ISWT), body weight (BW), knee extension strength (KES). Asterisks indicate significant for intragroup comparisons when *p* < 0.002 (0.05/25 comparisons = 0.002. The numbers of asterisks indicate the likelihood for the intragroup differences to be substantial, with 1 symbol (*) referring to likely differences, 2 symbols (**) to very likely, and 3 symbols (***) to almost certain differences. Threshold values for Cohen’s ES were trivial (0.0–0.19), small (0.20–0.59), moderate (0.60–1.19), large (1.20–1.99), and very large (>2.0). Numbers shown are the quantitative chances (%) that the effect is positive/trivial/negative. A substantial effect was defined as >75%. An adequate power analysis was defined as ≥0.8.

## Data Availability

The data presented are available in this study.

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
