# Peer review of "Feasibility, Safety, and Effects of an Aerobic Training Program with Blood Flow Restriction on Functional Capacity, and Symptomatology in Women with Fibromyalgia: A Pilot Study"

_biomedicines, 2024, doi:10.3390/biomedicines12081895_

Round 1

Reviewer 1 Report (Previous Reviewer 1)

Comments and Suggestions for Authors

Although the authors have made several changes to their submission of relatively small pilot study to assess the feasibility of aerobic training in female patients with fibromyalgia, overall these do not change the fact that there are major difficulties with the experimental design. These major difficulties cast doubt on the validity and applicability of the reported results. First, the study is very small, with seven patients versus a control group of six. Second, the study was not randomised. Third, it was not blinded.

As previously, I recommend that the authors consider setting up a much larger study which does not suffer from the above faults.

Author Response

Please read the attached pdf file. Thank you very much

Reviewer 2 Report (New Reviewer)

Comments and Suggestions for Authors

Dear Authors,

the paper is worth of interest and well-conducted. Some minor concerns are detailed below:

Abstract

-Methods. Detail the population. Report on statistics used.

Methods

-Interventions. Did the health carers use a manual treatment to limit the variability in the treatments administration? Were other treatments (e.g. drugs, manual therapy, injections, etc.) allowed during the interventions described?

-Describe possible effect modifiers, as well as potential sources of bias.

-Harms: who assessed them? and how?

Results

-Compliance rates to treatment: not reported.

Discussion

-Add a paragraph as for future research on the bio-psychosocial approach to fibromyalgia (including related outcome measures and recommended programs).

Comments on the Quality of English Language

 Minor editing of English language required

Author Response

Please read the attached pdf file. Thank you very much

Reviewer 3 Report (New Reviewer)

Comments and Suggestions for Authors

Title: Feasibility, safety, and effects of an aerobic training program with blood flow restriction on functional capacity, and symptomatology in women with fibromyalgia: A pilot study

In this study, fibromyalgia is considered a clinically significant topic because it is a chronic condition that predominantly affects women, causing widespread pain and extreme exhaustion. With all humility, these recommendations are collected with the intention that they can help to improve this work.

- Line 96: Please provide the calculated number of appropriate subjects for this study.

- What criteria were used to divide the subjects into Experimental Group 1 and 2, and how were the two groups divided?

- Line 200: Please provide a brief explanation of the measurement tools used in this study, along with their reliability and validity.

- Line 221: For the dynamometer (TKK 5401 221 Grip-D, Takei, Tokyo, Japan), please include the year of manufacture.

   Specification format for experimental equipment: model name, company name, country of manufacture, year of manufacture.

   Please apply the same format to all experimental equipment used.

- Line 279: Please provide the reason for conducting the experiment over 9 weeks and the basis for conducting aerobic walking twice a week.

   Did you monitor the subjects during aerobic walking with blood flow restriction?

- Line 305: Considering the small sample size (12 subjects, 6 in each group), why did you choose to use a 2-way ANOVA instead of non-parametric statistics?

- While there are many tables and results, organizing the main findings of this study would help readers understand the paper more easily.

- There were differences between groups in HR, FIQ morning tiredness, FIQ stiffness, MFI physical fatigue, CST, 6MWT, and Hand grip. What do you think are the reasons for these differences?

- What do you think are the main mechanisms of aerobic walking with blood flow restriction?

- Why do you think blood flow-restricted aerobic training may be feasible, safe, and more effective?

Author Response

Please read the attached pdf file. Thank you very much

Reviewer 4 Report (New Reviewer)

Comments and Suggestions for Authors

Presented pilot study aimed to to evaluate the feasibility, safety, and effects of an intervention based on a low-intensity, low-volume aerobic training program with  blood flow restriction with respect to functional capacity and the impact on symptomatology in women with fibromyalgia. The authors demonstrated that blood flow-restricted aerobic training may be feasible, safe, and more effective than unrestricted aerobic training as a physical exercise prescription tool to improve cardiorespiratory fitness, strength, and stiffness in women with fibromyalgia 

Great planned pilot study and well-written paper. All factors are described clearly. However, I have a small comment. In the introduction section please develop in short paragraph methods of treatment and rehabilitation of fibromyalgia patients in relation to current guidelines.

Congratulations to the authors!

Author Response

Please read the attached pdf file. Thank you very much

Reviewer 5 Report (New Reviewer)

Comments and Suggestions for Authors

This pilot trial evaluated the feasibility, safety, and effects of an aerobic training program with blood flow restriction for women with fibromyalgia. The premise of this study is relevant and important, however, there might be some points that could be significantly improved, and others need further clarification.

1.     The title is missing some of the study outcomes (i.e., hematological and biochemical effects of the interventions). A clear and informative title upfront with the representation of the study outcomes would help avoid confusion about the study goal and findings.

2.     Pg 1; Ln 25: “and 13 were selected by meeting all inclusion criteria”. How were the other participants selected? I cannot understand the mention of this information in this occasion of the manuscript. You may either need to elaborate more or omit this information from the abstract.

3.     Pg 1; Ln 25-26: I suggest specifying the blood flow restrictions (site of application, duration, and intensity).

4.     Moreover, the methods subsection of the abstract would benefit from a synopsis of aerobic training (type, training intensity, and duration).

5.     Pg 1; Ln 32-37: to strengthen the strength of the abstract, consider including relevant statistical values (p-values, confidence intervals, effect sizes) alongside the reported findings.

6.     Pg 2: Ln 68-76: authors may need to provide more context and background information about the role of aerobic training for patients with fibromyalgia in light of the existing literature and highlight how important it is to combine the blood flow restriction with aerobic training for such a patient population.

7.     Pg 2: Ln 79-81: you stated, “Elements to highlight in aerobic training program with blood flow restriction focused on fibromyalgia could include the ability to considerably reduce the volume and intensity of the applied training load necessary to produce adaptations”. Examples of these adaptations are worthy of mention.

8.     Pg 2: Ln 81-87: you made reference to the effect of occlusive training in older adults, overweight people, and others with chronic conditions. Are there any previous studies that explored the impact of this intervention in patients with fibromyalgia?  

9.     Pg 3: Ln 126-130: I remain unclear on how the group assignment was performed. To ensure a complete understanding of the methodology, I would appreciate it if you could elaborate on the group assignment process.

10.  I have a major concern regarding the lack of a control group that received no treatment (i.e. neither aerobic training nor blood flow restriction). This makes it difficult to draw a cause-and-effect relationship between the intervention and the observed outcomes, weakens the overall credibility of the study, and limits the usefulness and applicability of its findings.

11.  Is the study powered enough for the design and outcome measures? I am not sure. Data from just 13 participants is likely insufficient, it would be interesting if the authors clarified the settings they adopted for the sample size calculation.

12.  Calculating the sample size in pilot studies is quite important, although it serves different purposes than in the main clinical trial. A sufficient sample size allows for more precise estimates of the effect size and other key parameters that are crucial for designing the main trial effectively.  

13.  Pg 5; Table 1: The table should demonstrate whether a statistical difference exists between the intervention and non-intervention groups.

14.  Pg 5; Table 1: which group “compliance” and “non-compliance” data belong?

15.  Pg 5; Ln 213: Several tests have been used to assess cardiopulmonary fitness. What aspects of fitness is each test intended to assess?

16.  Pg 7; Ln 294-303: the site of application of BFR should be clarified.

17.  The methodology used to determine the intensity of aerobic training is not the most reliable method. I would suggest considering more reliable methods like HRmax or VO2max in the main trial if you are planning for a large-scale study.

18.  Pg 7: Ln 313-314: “adjusting for multiple comparisons using the Bonferroni method by dividing the 0.050 significance level by the number of comparisons”. Is there a need for this analysis in 2x2 mixed-model ANOVA? If yes, why? and what was the cutoff for statistical significance in this case?

19.  Have you decided the difference between groups using the group-by-time interaction effect of the main effect?

20.  Tables 2a, 2b, 3, 4, and 5: what do mean by standardized difference? Do you mean the average pre-to-post-change difference between the study groups?

21.  What qualitative assessments in the above-mentioned tables refer to?

22.  Still, I could understand the significance of the post-hoc analysis.

23.  Tables numbering needs revisions. I can see different tables with the same numbers.

24.  Pg 24; Ln 591: I suggest reminding the readers of the primary focus of the study at the beginning of the discussion.

25.  Also, the summary of the results should explicitly address whether they support or contradict the initial hypothesis.

26.  The rest of the discussion reads well. Authors interpreted the study findings, tried to compare them with previous reports (as possible), highlighted the study merits, and acknowledged the study limitations

27.  Pg 25; Ln 781: The first 3 points in the practical applications are not supported by the findings of the current study. The focus was not to assess the validity of these measures in patients with fibromyalgia.

28.  Citing 92 references is too much for a pilot study. Consider reducing this number, keeping the most relevant and supportive citations only.

Author Response

Please read the attached pdf file. Thank you very much

Round 2

Reviewer 1 Report (Previous Reviewer 1)

Comments and Suggestions for Authors

I thank the authors for their response to my latest comments. The authors imply that the primary purpose of their research study was in relation to safety/feasibility. This, according to them, explains the experimental design. To quote from the start of their latest response: "I would like to point out that the experimental design was carried out under safety and feasibility criteria ..."

Interestingly, safety is not mentioned at all as an aim of this study in their original trial registration with ClinicalTrials.gov. Their original title does not include mention of safety/feasibility. Their detailed description of their trial when registering it states: "Intervention study in people with fibromyalgia with the implementation of a lower body occlusive tool in aerobic exercise. The study aims to descriptively test the impact on quality of life and functional autonomy of occlusive training in a controlled and individualized way in two groups: group 1 aerobic exercise with use of occlusive tool and group 2, aerobic exercise without occlusive tool in a period of 9 weeks twice a week." Again, there is no mention of safety. Their outcome measures (CoQ10, TAS, WPI, etc.) in the original trial registration documentation are stated in a manner which suggests that the authors were expecting to find differences in the intervention group.

I accept what the authors say about blinding. However, my overall decision remains unchanged.

Author Response

Please see attached file. Thank you very much.

Reviewer 3 Report (New Reviewer)

Comments and Suggestions for Authors

Thank you for your efforts in thoroughly revising the reviewer's comments. The topic of this study, fibromyalgia, is clinically important; however, as an experimental study with only 13 subjects, 6 in each group, the sample size is too small, which is a major limitation. How can we overcome this issue?

Author Response

Please see attached file. Thank you very much.

Reviewer 5 Report (New Reviewer)

Comments and Suggestions for Authors

The authors attentively addressed the previous comments and suggestions, resulting in a significant improvement of the manuscript. With the implementation of a few minor corrections, the work would be publishable.

Pg 1; Ln 27. To ensure clarity, could you be more specific about the location within the lower body you have in mind?

Is there a reference for the application of 14 minutes of blood flow restriction (divided into two periods; each 7 minutes with a 3-minute interval)?

Pg 1; Ln 30. What was the duration of aerobic training in each session?

Pg 1; Ln 42. I think “IC 95% and statistical power > 80” should be deleted.

Round 3

Reviewer 3 Report (New Reviewer)

Comments and Suggestions for Authors

Thank you for your efforts in thoroughly revising the reviewer's comments. Accept in present form

This manuscript is a resubmission of an earlier submission. The following is a list of the peer review reports and author responses from that submission.

Round 1

Reviewer 1 Report

Comments and Suggestions for Authors

The authors present a pilot study to assess the feasibility of aerobic training in female patients with fibromyalgia.

Unfortunately, there are major difficulties with the experimental design, which cast doubt on the validity and applicability of the reported results. First, the study is very small, with seven patients versus a control group of six. Second, the study was not randomised. Third, it was not blinded.

I recommend that the authors consider setting up a much larger study which does not suffer from the above faults.

Comments on the Quality of English Language

Some slight errors in the language.

Reviewer 2 Report

Comments and Suggestions for Authors

Dear Authors

Thanks a lot for the opportunity you have offered me to revise the fascinating manuscript " Feasibility, safety, and effects of an aerobic training program with blood flow restriction on functional capacity, lipid profile, and symptomatology in women with fibromyalgia: A pilot study. ".

As a significant strength, this manuscript evaluates the feasibility, safety, and 24 effects of this tool in women with fibromyalgia. This proposal is novel and adds to existing literature. The paper is really nice!

As a major weakness, The manuscript would benefit from additional details and clarity regarding methodological steps to improve understanding and quality of work. Therefore, I suggested strategies to enhance the authors' reporting.

Overall, my peer review is a minor revision. I think the paper will be published soon after the amelioration.

¶MAJOR ISSUES:

#INTRODUCTION:

*rationale The authors should emphasize the novelty of their study and provide evidence from existing literature.

*background: small fibre neuropathy has been demonstrated in fibromyalgia (PMID: 33561533 doi: 10.1016/j.jbspin.2021.105153). In agreement, I suggest the authors mention this in the background.

*main questions: I suggest that authors provide a clearer explanation of the reasoning behind their question.

#METHODS:

*TREND: The use of TREND is nice. However, I suggest that the authors add the guidelines as a supplementary file.

*hospital: was It private or public? Specify.

*inclusion and exclusion criteria should be based on evidence about fibromyalgia. Can you do it?

*assessment: who performed the assessment? Is there a blind assessment?

*feasibility: please add the appropriate references for its parameters. If it was a discretionary choice, please report it as a limitation within the discussion.

*safety: please add the appropriate references for the parameters. If it was a discretionary choice, please report it as a limitation within the discussion.

*functional capacity: please add the appropriate references for the parameters. If it was a discretionary choice, please report it as a limitation within the discussion.

#RESULTS:

*Reporting: I suggest authors organize subchapters in their results.

#DISCUSSION:

*limitation section: the limitation section is missing. Please report it in the discussion, mentioning, for example, the low sample, the mono-centric nature of the study, the lack of randomization, etc. This prevents you from possible criticisms.

*Implications for research: I suggest authors report the need for future randomized controlled trials to assess the efficacy of the interventions.

*Implications for clinical practice: I suggest authors suggest the incorporation of the intervention during the administration of functional exercises with an external focus of attention (PMID: 28286760  doi: 10.1155/2017/2946465). This will improve the quality of your discussion.

¶MINOR ISSUES:

#ENGLISH:

*I suggest the authors have the English revised by a native speaker as there are several typos and inaccuracies (e.g., i.e.).

#GENERAL

*Acronyms. I recommend that you provide a complete report of all the acronyms mentioned throughout the manuscript.

#REFERENCES:

*The authors should consider referencing recent, high-quality studies, such as systematic reviews and meta-analyses, to improve the paper's quality.

#TABLES and IMAGES

*Acronyms. The manuscript should include full reporting of all acronyms used.

#ABSTRACT

*Acronyms. The abstract should include full reporting of all acronyms used (e.g., 6MWT, ISWT, 30CST, TUGT, etc).

Comments on the Quality of English Language

minor english edits are needed.

Round 2

Reviewer 1 Report

Comments and Suggestions for Authors

I thank the authors for kindly addressing my previous comments. I am glad to see that the analyses were carried out in a blinded fashion. However, it remains the case that this study was very small and not randomised.